# Gene Expression Profiles and microRNA Regulation Networks in Tiller Primordia, Stem Tips, and Young Spikes of Wheat Guomai 301

**DOI:** 10.3390/genes10090686

**Published:** 2019-09-06

**Authors:** Junchang Li, Zhixin Jiao, Ruishi He, Yulong Sun, Qiaoqiao Xu, Jing Zhang, Yumei Jiang, Qiaoyun Li, Jishan Niu

**Affiliations:** National Centre of Engineering and Technological Research for Wheat/Key Laboratory of Physiological Ecology and Genetic Improvement of Food Crops in Henan Province, Henan Agricultural University, Zhengzhou 450046, China

**Keywords:** wheat (*Triticum aestivum* L.), tissue, RNA-seq, differentiation, gene co-expression, molecular regulation

## Abstract

Tillering and spike differentiation are two key events for wheat (*Triticum aestivum* L.). A study on the transcriptomes and microRNA group profiles of wheat at the two key developmental stages will bring insight into the molecular regulation mechanisms. Guomai 301 is a representative excellent new high yield wheat cultivar in the Henan province in China. The transcriptomes and microRNA (miRNA) groups of tiller primordia (TPs), stem tips (STs), and young spikes (YSs) in Guomai 301 were compared to each other. A total of 1741 tillering specifically expressed and 281 early spikes differentiating specifically expressed differentially expressed genes (DEGs) were identified. Six major expression profile clusters of tissue-specific DEGs for the three tissues were classified by gene co-expression analysis using *K*-means cluster. The ribosome (ko03010), photosynthesis-antenna proteins (ko00196), and plant hormone signal transduction (ko04075) were the main metabolic pathways in TPs, STs, and YSs, respectively. Similarly, 67 TP specifically expressed and 19 YS specifically expressed differentially expressed miRNAs were identified, 65 of them were novel. The roles of 3 well known miRNAs, tae-miR156, tae-miR164, and tae-miR167a, in post-transcriptional regulation were similar to that of other researches. There were 651 significant negative miRNA–mRNA interaction pairs in TPs and YSs, involving 63 differentially expressed miRNAs (fold change > 4) and 416 differentially expressed mRNAs. Among them 12 key known miRNAs and 16 novel miRNAs were further analyzed, and miRNA–mRNA regulatory networks during tillering and early spike differentiating were established.

## 1. Introduction

Wheat (*Triticum aestivum* L.) is one of the most important cereal crops around the world with wheat grain yield being the major breeding trait. There are many factors affecting the final yield of wheat, among which, spike number per hectare and thousand-grain weight directly determine the wheat grain yield. Therefore, tiller ability and spike traits are very important for wheat. A certain amount of tillers is very important for the accumulation of cereal organism mass during the vegetative growth stage [1]. Wheat stem elongation occurs from terminal spikelet initiation to anthesis, and it determines the number of fertile florets [2]. Wheat spike development begins with the transition of the meristem from the vegetative state to the reproductive state and ends when the plant matures [3]. Early crop inflorescence differentiation significantly and directly influences the subsequent floral development, grain number per spike, thousand-grain weight and final yield [4,5]. Such wheat tillering and spike differentiation should be carefully studied.

The molecular genetic mechanism of tillering is one of the major issues in crop science. There are many crop tiller-related genes and microRNA (miRNAs) have been reported. Four wheat tiller inhibition lines or mutants are controlled by a single nuclear gene [6], the gene *tin1* is on chromosome arm 1AS [7], *tin2* is on chromosome 2A [8], *tin3* is on chromosome arm 3AL [9], and *ftin* is on chromosome arm 1AS [10]. A major quantitative trait locus conferring high tillering ability is mapped on chromosome arm 2DS in wheat [11]. In rice (*Oryza sativa* L.), *MONOCULM 1* (*MOC1*) is the first investigated gene controlling tillering [12]. *MOC1* encodes a GRAS (GAI, RGA, and SCR) family nuclear protein and promotes outgrowth of axillary buds [13,14]. Rice *LAX1* (*lax1, lax panicle 1*) regulates mRNA expression and subsequently controls protein trafficking. *LAX1* is a regulator controlling axillary meristem initiation during reproductive development [14,15]. Branching mutants of more axillary growth (*max*) of Arabidopsis (*Arabidopsis thaliana*) and dwarf (*d*) of rice show increased branching and reduced stature relative to wild-type plants [16]. A number of rice mutants conferring an increased number of tillers have been reported, they are *MAX2* [17], *htd1* [18], *MAX3* [19], *MT1* [20], *D14* [21], and *D27* [22]. In barley (*Hordeum vulgare* L.), lower expression levels of *HvD27*, *HvMAX1*, *HvCCD7*, and *HvCCD8* indicate that ABA suppresses strigolactones biosynthesis, leading to enhanced tiller formation [23].

Inflorescence development is another focus in crop science. Most molecular genetic reports about inflorescence development are from Arabidopsis. In rice and maize (*Zea may* L.), multiple genes regulate inflorescence development together [24]. To date, many kinds of mutants related to spike development have been reported in wheat [25,26,27,28]. However, the molecular mechanism of wheat spike development is largely unknown. The famous *Q* gene controls threshability, rachis fragility, and spike shape traits [29]. *APETALA2* (*AP2*)-like gene co-segregated with *Q* is known to play a major role in controlling floral homeotic gene expression, which is considered as an excellent candidate for *Q* [30]. At the late developmental stage of inflorescence, anther/ovary size share closer connections with grain number/size traits at the late vs. early stages of floral development [31]. For spike morphology, some traits (e.g. total and fertile spikelet number, spike length) display high prediction abilities, the change of these traits helps to increase the yield of wheat [32]. A durum wheat (*Triticum turgidum ssp*. *durum*) *ARGONAUTE1d* (*AGO1d*) gene controls spike length and grain number per spike [3], but its role in wheat spike structure formation remains unclear. The locus *Grain Number Increase 1* (*GNI1*) is an important contributor to floret fertility. The reduced-function allele *GNI-A1* contributes to the increased number of fertile florets per spikelet [33]. Floradur-*bht-A1-NILs* are essential for increasing grain number. Understanding the molecular and genetic mechanism of the source–sink interaction is important for a clearer picture of the complex signaling network regulating sink strength and source activities in *Triticum* family [34]. Genetic mutations of wheat *Ppd1* (*Photoperiod-1*) and *WFZP-A* (wheat *FZP, FRIZZY PANICLE*) genes lead to abnormal spike differentiation [35,36]. It is found that the genetic factors and hormones play important roles in regulating the transition from vegetative to reproductive [37]. Although these studies have discovered some molecular mechanisms of the plant inflorescence development, little is known about the main genetic factors controlling the wheat spike differentiation. 

It is clear the plant miRNAs play central roles in switching from vegetative to reproductive growth. Some miRNA regulations show obvious tissue specificity. For example, miR172 regulates the development of floral organs by targeting *AP2*-like genes in maize and rice [38,39,40], but it does not affect tillering in rice [40]. Overexpression of miR156 produces more lateral roots, whereas reducing expression of miR156 leads to fewer lateral roots in Arabidopsis [41]. The miR156 target genes of the *SQUAMOSA PROMOTER BINDING PROTEIN-LIKE* (*SPL*) family promote plant growth and development in wheat including increasing tiller numbers and influencing spikelet formation. The miR156 is involved in a large regulatory network [41,42,43,44,45,46,47,48,49]. The miR159, miR167, and miR319 interactively target important transcription factors to regulate floral organogenesis, including MYBs, TCPs, SBPs, and ARFs [50,51,52]. Although it is clear that these miRNAs play pivotal roles in meristem activities, the functions of most wheat miRNAs during tillering and spike differentiating are largely unknown.

It is valuable for wheat molecular mechanism study and molecular breeding to discover the gene expression profiles and miRNA regulation networks during tillering and spike differentiating. Henan is the most important wheat production province in China. In recent years, the annual wheat growth area is more than 5.33 × 10^6^ hectares in Henan, and the output is about 1/4 of the wheat grain product in China. Guomai 301, bred in our lab, is a representative excellent new high yield wheat cultivar in Henan province. A study on the gene expression profiles and miRNA regulation networks of Guomai 301 during tillering and spike differentiating will provide useful information for wheat breeding and production in China, as well as in other wheat growth areas all over the world. RNA sequencing is a widely used powerful technique in the studies of normal growth and development of various crops [53,54,55], which was also employed in this study. The miRNA and mRNA co-expression profiles, the possible miRNA target candidate genes, and the comprehensive post-transcriptional regulation networks of Guomai 301 during wheat tillering and spike differentiating were reported here.

## 2. Materials and Methods 

### 2.1. Plant Materials and Growth Conditions

Guomai 301 is a powdery mildew resistant, high yield wheat cultivar bred in our lab, the National Centre of Engineering and Technological Research for Wheat, Henan, P. R. China [56]. In 2016–2017 wheat growth seasons, Guomai 301 was planted in our experimental field at Houwang Village, Xingyang City, Henan, P. R. China (34°25′ N, 115°39′ E, 49 m a.s.l.). The seeds were sown in plots of 3.0 m in length and 2 m in width, the distance between rows was 0.25 m, and 30 seeds were planted in each row. Fertilizer and weed management were similar to wheat breeding [57].

### 2.2. Morphology Observation

Tiller primordia (TPs),young spikes (YSs) and stem tips (STs) of Guomai 301 were observed from the early developmental stage [5,58] with an inverted microscope (SRL-7045A, Beijing Century Science Letter Scientific Instruments Co. Ltd. Beijing, China). The out leaves and sheaths were removed to show the TPs and YSs with an anatomical needle. The images were captured by a camera (Nikon Coolpix 4500, Nikon Corporation, Tokyo, Japan). The big tillers and big spikes at adult plant stage were observed by the naked eye.

### 2.3. Sample Preparation and RNA Extraction

In November 2016, when Guomai 301 developed at the 3-leaf to 4-leaf stage, the TPs were dissected out, frozen immediately in liquid nitrogen and stored at –80 ℃. In March 2017, when spikes of Guomai 301 developed at about spike differentiation stage 5 [5], the YSs and STs (Figure 1) were dissected out, frozen immediately in liquid nitrogen and stored at –80 ℃. Three super bulk samples of TPs, STs, and YSs were prepared with three biological replicates. Each super bulk of TP included more than 20 independent individuals. Each super bulk of ST and YS included more than 100 independent individuals. 

The total RNAs were extracted majorly as our report [6] using TRIZOL reagent (TransGen Biotech, Beijing, China). DNA was removed using DNase (Invitrogen, Shanghai, China). RNA concentration was measured using NanoDrop 2000 (NanoDrop Technologies, Wilmington, DE, USA). RNA integrity was assessed using the RNA Nano 6000 Assay Kit of the Agilent Bioanalyzer 2100 system (Agilent Technologies, CA, USA). 

### 2.4. Transcriptome Sequencing and Data Analysis 

Nine mRNA libraries were constructed using NEBNext®Ultra™ RNA Library Preparation Kit for Illumina (Illumina, San Diego, CA, USA) following manufacturer’s recommendations. Transcriptome was paired-end sequenced with the Illumina HiSeq Xten platform in Biomarker Biotechnology Corporation (Beijing, China), which generated approximately 125 bp paired-end (PE) raw reads by LC Sciences (Houston, TX, USA). The clean reads from the nine transcriptomes were obtained from raw data by filtering out adaptor sequences and low-quality reads, then the remaining clean reads were assembled using Trinity software for de novo transcriptome assembly with a reference genome (Ensemble Triticum_aestivum. TGACv1: ftp://ftp.ensemblgenomes.org/pub/plants/release-32/fasta/triticum_aestivum).

The annotation of the transcriptome sequences was carried out based on their homologous sequences obtained from BLAST [59] search against the public databases, including the Non-Redundant Protein database (http://www.ncbi.nlm.nih.gov), the Swiss-Prot database (http://www.uniprot.org), the Gene Ontology database (http://www.geneontology.org), Cluster of Ortholog Genes database (http://www.ncbi.nlm.nih.gov/COG), the Eukaryotic Orthologous Groups database (http://www.ncbi.nlm.nih.gov/KOG), Protein Family Database (http://pfam.xfam.org) and the Kyoto Encyclopedia of Genes and Genomes database (http://www.genome.jp/kegg). The involved biological pathways of the new genes referring to KEGG database were analyzed using KOBAS2.0 [60]. The amino acid sequences of the new genes were predicted and annotated using HMMER [61], referred to the several databases. Quantitative gene expression was calculated with the method of fragments per kilobase of transcript per million mapped fragments (FPKM) [62]. The bioproject accession of the transcriptome data in NCBI is PRJNA551725.

Pairwise difference analysis was conducted on gene expression levels among the three different tissues using DESeq R packages [63]. The parameters of the fold change (FC, Log_2_FC ≥ 1 or ≤ −1) and false discovery rate (FDR ≤ 0.01) were used to identify DEGs. The FCs indicating the ratio of the mRNA numbers were detected in the two analyzed samples. The FDR was corrected by the *p*-value of the significant difference. If FPKM value of any one transcript of a gene in each of the three sample replicates was more than one, we defined it was a valid gene. If the average value of FPKM of a gene in one tissue was more than twice that of the other two tissues, we said it was up-regulated against the others. We defined such a gene as a tissue-specific gene. In another words, these genes were expressed at all three tissues but were particularly highly expressed in one of them. Pearson correlation coefficient (PCC) and principal component analysis (PCA) were carried out to evaluate the indices of biological repetition correlation [64,65]. Log_2_FC ≥ 2 or ≤ −2 were chosen as the threshold parameter to screen for DEGs, and the average value of the three repetitions of each tissue was taken to perform gene co-expression analysis by *K*-means clustering algorithm [66].

### 2.5. miRNA Sequencing and Data Analyses

The nine small RNA libraries were constructed using the same samples as those for transcriptome sequencing described above. Each reaction solution was 6 μL containing 1.5 μg total RNA. The libraries were constructed using the Small RNA Sample Preparation kit (Illumina, San Diego, CA, USA) according to the instructions. The clean reads were obtained by filtering low-quality reads and adaptor contaminants, and BLAST in Silva (https://www.arb-silva.de), GtRNAdb (http://gtrnadb.ucsc.edu), Rfam (http://rfam.xfam.org), and Repbase (https://www.girinst.org/repbase) databases using Bowtie software (http://bowtieapp.com); [67]. The unannotated reads were obtained by discarding the reads matching known rRNA, tRNA, snRNA, ncRNAs, snoRNA, and repeat sequences. Sequence alignment and subsequent analysis were performed using Bowtie software with the designated TGACv1 as a reference genome (Ensemble *Triticum_aestivum*. TGACv1: ftp://ftp.ensemblgenomes.org/pub/plants/release-32/fasta/triticum_aestivum), For Bowtie, 0 (full length mismatch) was used as mapping parameter, and the positional information of the unannotated reads on the reference genome was predicted. The miRNAs identical to the known miRNAs in the mature miRNA database miRBase (Release21, http://www.mirbase.org) were considered as known miRNAs. The miRDeep2 [68] was used to predict unknown or novel miRNAs. Family analysis of the detected miRNAs and novel miRNAs were performed based on the similarity among miRNA sequences. Transcripts per million (TPM) algorithm was used to normalize the expression levels of miRNAs [69]. The bioproject accession of the miRNA-seq data is PRJNA551738.

The DE miRNAs between samples were identified by using a Student’s *t*-test based on the experimental design. In this test, the significance threshold was set at 0.05. Similarly, PCC and PCA were chosen to evaluate biological repetition correlation. 

### 2.6. Prediction of the miRNA Target Genes 

Comprehensive analysis was performed between the transcriptome data and miRNA data. TargetFinder software (http://www.bioinformatics.org/mirfinder) was used to predict the targeted genes of miRNAs [70]. Based on plant target penalty strategy, it is widely used in plant miRNA target gene prediction. Briefly, miRNA sequences matched to the reference mRNA sequences and potential targets were computationally predicted by the match/mismatch-scoring ratio. Mismatched pairs were scored as 1, and G:U pairs were scored as 0.5. If the mismatches were located between the second base and the thirteenth base counting from the 5’ end of the miRNA sequence the score was doubled. Only predicted targets with scores less than four were considered reasonable.

### 2.7. qRT-PCR

The samples TPs, STs and YSs of Guomai 301 were prepared at 5 time points for real-time PCR. The intervals of the TP samples were seven days from 30 December 2016 to 27 January 2017. The intervals of the ST and YS samples were seven days from 1th March 2017 to 29 March 2017. Among them, the samples prepared at the first time point were used for RNA-seq. All primers were designed using Primer Premier 5.0 (http://www.premierbiosoft.com/primerdesign/index.html). The information of primers was listed in Appendix A. Reverse transcription was performed using TransScript^®^ All-in-One First-Strand cDNA Synthesis SuperMix for qPCR (TransGen Biotech, Beijing, China). Real time - quantitative reverse transcription PCR (qRT-PCR) was performed using TransStart^®^ Top Green Qpcr SuperMix (2×) (TransGen Biotech, Beijing, China) according to the manufacturer’s protocol on the CFX Connect^TM^ Real-Time System (Bio-Rad, Hercules, CA, USA). The ingredients of the reaction system were strictly carried out according to the instructions. The wheat actin gene was used as an internal control gene. The qRT-PCR reactions were performed in 20 µL volumes [6]. The gene expression levels were calculated according to the 2^−ΔΔCT^ method [71].

## 3. Results

### 3.1. Tillering and Spike Differentiating Traits of Guomai 301

Guomai 301 is a representative semi-winter wheat cultivar in Henan, China. It has dark green leaves, thick stems, long awns, large spindle-shaped spikes, and an average 37.4 grains per spike (Figure 1A). The average yields in 2010, 2011, and 2012 were 8068.5, 8601, and 8034 kg·hm^−2^, respectively. The typical traits of Guomai 301 are medium tiller number, high percentage of ear-bearing tillers and good seed setting ability, which guarantee its high yield potential. Guomai 301 has few ineffective tillers, and its average tiller number is 21.73 when it being sparsely sown [6], and the effective tiller number for optimized highest yield per ha is about 2.5. The spike differentiation of Guomai 301 is relatively slower compared with other cultivars in Henan. The phenotypes of the tiller primordia at the early tillering stage (Figure 1B,C) and young spikes (Figure 1D–M) at early spike differentiation stages are showed in Figure 1. Tiller primordia (Figure 1B) at the early tillering stage, stem tips and young spikes (Figure 1F) at glume primordia visible stage were sampled for mRNA and micro-RNA transcriptomic analyses.

### 3.2. Overview of Transcriptome Sequencing Data

Nine libraries (TPs: T0a, T0b, T0c; STs: T1a, T1b, T1c; YSs: T2a, T2b, T2c) were sequenced, and 79955, 81241, 82379, 75994, 77460, 76872, 76502, 75073, and 75160 genes were identified, respectively. A total of 103.82 Gb clean bases were obtained, and the clean bases of each sample reached 10.29 Gb. The average Q30 percentage was more than 88.84% (Appendix A). 12,417 new genes were identified and named as Wheat_newGene_1 - Wheat_newGene_12417, of which 8829 were functional annotated referring to various databases (Appendix A). The reads were compared with the *T. aestivum* reference genome. The comparison efficiency of each sample was between 79.23% and 82.46%. The average percentage of unique mapped reads was 70.88% (Appendix A). The average number of detected genes in the nine libraries (FPKM ≥ 1) was close to 45000, and there was no significant difference among the three tissues (Figure 2C). A total of 41706 valid genes (72.20%) were found in the three tissues. The numbers of genes expressed only in T0, T1, and T2 were 5492 (9.52%), 1167 (2.02%) and 2147 (3.72%), respectively (Figure 2A). Obviously, tiller primordia have the most tissue-specific genes during tillering at the early vegetative growth stage, and the expressed genes involving in cell differentiation and vital movement in tiller primordia were more than in other tissues.

The three-dimensional map of PCA clearly showed the correlation between transcriptome sequencing samples (Appendix A). The points in different colors in PCA represented different samples, the top two eigenvectors (31.30% and 38.98%) clearly separated these samples (Appendix A), which was consistent with the morphological pattern of the different tissues in wheat. PCC analysis showed each correlation coefficient was more than 0.908 between all replicated samples (Appendix A), which demonstrated the biological replicates were highly consistent. 

### 3.3. Overview of miRNA Sequencing Data

A total of 420 miRNAs were identified, and 414, 416, 419, 397, 400, 400, 390, 398, and 386 (Appendix A) in the nine samples (tps: S0a, S0b, S0c; sts: S1a, S1b, S1c; yss: S2a, S2b, S2c; the samples sps, sts and yss were prepared from the same original samples of TPs, STs, sand YSs), respectively. Through quality control, we removed low-quality reads and corrupted adapter sequences (reads < 18 nt or > 30 nt long). The clean read number of each sample was more than 15.32 M (Appendix A). The average Q30 percentage was more than 98.42%. The reads were compared with the *T. aestivum* reference genome. The mapping efficiency of each sample was between 52.64% and 67.91% (Appendix A).

The length distributions of miRNAs were similar among different samples. The most miRNAs were 21 nt (42.62%) and 24 nt (37.14%) long (Appendix A). In total, we identified 77 known miRNAs belonging to 34 miRNA families, and 343 predicted novel miRNAs (Appendix A).

The three-dimensional map of PCA clearly showed the correlation between miRNome sequencing samples (Appendix A). The top two eigenvectors (31.76% and 21.78%) of PCA clearly showed the correlation between samples (Appendix A) and separated the nine samples. PCC analysis showed every correlation coefficient was more than 0.919 between replicated samples (Appendix A), which demonstrated the biological replicates were highly consistent. 

### 3.4. DEGs between TP and YS

To further explore wheat tillering related and spike differentiation related biological processes or pathways, a total of 11,668 DEGs were identified between tiller primordia and young spikes (Figure 2B). Among them, 8685 genes expressed at a lower level, and 2983 genes were highly expressed in young spikes compared to tiller primordia. According to the percentage of DEGs in all genes, these DEGs were mainly classified into 44 of the 54 subcategories in the Gene Ontology (GO) database (biological process, cellular component, and molecular function) (Figure 3). A total of 2351 DEGs were assigned to 123 pathways referring to KEGG database, the top enriched pathways were obtained (Appendix A), including phenylpropanoid biosynthesis (ko00940), phenylalanine metabolism (ko00360), and starch and sucrose metabolism (ko00500). This suggests that these common metabolic pathways are extremely active during the growth transition.

Compared to the tiller primordia, the up-regulated genes in young spikes were separated referring to KEGG database. The top ten enriched pathways (Figure 4) were homologous recombination (ko03440), ubiquitin mediated proteolysis (ko04120), spliceosome (ko03040), plant hormone signal transduction (ko04075), mRNA surveillance pathway (ko03015), purine metabolism (ko00230), mismatch repair (ko03430), carbon metabolism (ko01200), starch and sucrose metabolism (ko00500), pyrimidine metabolism (ko00240), RNA transport (ko03013), nucleotide excision repair (ko03420), and plant-pathogen interaction (ko04626). 

The up-regulated genes in tiller primordia that involved top ten enriched pathways (Figure 4) were phenylpropanoid biosynthesis (ko00940), phenylalanine metabolism (ko00360), starch and sucrose metabolism (ko00500), biosynthesis of amino acids (ko01230), carbon metabolism (ko01200), plant hormone signal transduction (ko04075), glutathione metabolism (ko00480), ribosome (ko03010), amino sugar and nucleotide sugar metabolism (ko00520), cyanoamino acid metabolism (ko00460), cysteine and methionine metabolism (ko00270). 

### 3.5. Tissue-Specific Genes in Tiller Primordia

A total of 1741 genes (Appendix A) were considered to be specifically expressed in TPs. According to gene number and *p*-value of main terms in GO database, the top ten terms of gene enrichment were listed (Appendix A). They were apoplast (GO:0048046), plant type cell wall (GO:0009505) (belong to cellular component), peroxidase activity (GO:0004601), transporter activity (GO:0005215), hydrolase activity (GO:0004553) (belonging to molecular function), transcription from plastid promoter (GO:0042793), protein targeting to chloroplast (GO:0045036), chloroplast organization (GO:0009658), lignin biosynthetic process (GO:0009809), and chloroplast organization (GO:0009658) (belonging to biological process).

The major top metabolic pathways (Table 1) were classified based on the annotations in KEGG database. These genes involved pathways were ribosome (ko03010), biosynthesis of amino acids (ko01230), and nitrogen metabolism (ko00910). The genes involved in ribosomal metabolic pathways were the most significant. 

### 3.6. Tissue-Specific Genes in Young Spikes

A total of 281 genes were considered to be specifically expressed in Young spikes (Appendix A). According to gene number and *p*-value of main terms in GO database, the top ten terms of gene enrichment were listed (Appendix A). They were nucleus (GO:0005634) (belonging to cellular component), protein dimerization activity (GO:0046983), DNA binding (GO:0003677) (belonging to molecular function), meristem development (GO:0048507), flower development (GO:0009908), floral meristem determinacy (GO:0010582), leaf vascular tissue pattern formation (GO:0010305), regulation of transcription and DNA-templated (GO:0006355), auxin activated signaling pathway (GO:0009734), meristem development (GO:0048507) (belonging to biological process).

The metabolic pathways (Table 1) associated with young spike differentiation (YSs) included plant hormone signal transduction (ko04075) and DNA replication (ko03030).

### 3.7. DEG Co-Expression Clusters 

In order to find out tissue type-specific gene expression trends from DEGs among the three tissues, a gene co-expression analysis was performed using *k*-means cluster, setting *k* as 6, and normalizing values of the three biological replicates for each tissue. A total of 8310 genes (Appendix A) were selected and classified into six groups (Figure 5A,B). 

These clusters implied that a variety of genetic modules were related to their functions. Different DEG modules were the embodiments of their functions. To identify the tissue-specific biological processes different among the three samples, the functions of DEGs in every cluster were classified referring to GO database. Many specifically enriched terms were found. Here, we listed the main biological processes of the six clusters in Figure 5C (*p* < 0.05). 5869 genes in clusters K1 and K2 were up-regulated in tiller primordia that commonly regulate wheat growth. Genes involving cell wall modification (GO:0042545) were greatly enriched in cluster K1, meanwhile, biological processes including response to karrikin (GO:0080167), defense response to fungus (GO:0050832), lignin biosynthetic process (GO:0009809), and plant-type secondary cell wall biogenesis (GO:0009834) were greatly enriched. 

844 genes in clusters K3 and K5 were specifically expressed in stem tips. These genes were mainly involved in photosynthesis-light harvesting (GO:0009765), response to red light (GO:0010114) (belonging to Cluster K5). 

Similarly, 1597 genes in clusters K4 and K6 had the highest expression levels in young spikes. These genes mainly regulated wheat floral organ differentiation and development, including meristem development (GO:0048507), flower development (GO:0009908), apical protein localization (GO:0045176), and maintenance of inflorescence meristem identity (GO:0010077) (belonging to Cluster K6).

To further explore the metabolic pathways which these clustered genes participated in, various metabolic pathways were identified referring to KEGG database (Appendix A). For the six clusters, we found out that most genes were clearly involved in starch and sucrose metabolism (ko00500) in wheat tiller primordia. In addition, the genes involved in photosynthesis-antenna proteins (ko00196) and plant hormone signal transduction (ko04075) were highly expressed in stem tips and young spikes, respectively. At the same time, plant hormone signal transduction (ko04075) appeared in the significant metabolic pathways of the three tissues (Cluster K1, K3, and K6). It showed that different plant hormones played indispensable roles in different tissues or periods. These genes involved in plant hormone signal transduction pathways were listed in Appendix A. Various genes encoding auxin responsive proteins were found in different tissues, homologs of *IAA12, IAA14, IAA27*, and *IAA31* were found highly expressed in tiller primordia, *IAA27, IAA30* were found highly expressed in stem tips, and *IAA11* was up-regulated in young spikes. In addition, various protein kinases and transcription factors (TFs) in plant hormone signal transduction pathways played important roles in different tissues, too.

### 3.8. DE miRNAs Between Tiller Primordia and Young Spikes

Here, we focused on the miRNAs in tiller primordia and young spikes involving in the conversion of vegetative growth to reproductive growth. Compared the miRNAs in tiller primordia and young spikes each other, 97 miRNAs were significantly highly expressed in tiller primordia, while 64 miRNAs were significantly highly expressed in young spikes (*p* < 0.05, Appendix A). For those known miRNAs, the pairwise analysis showed that tae-miR156, tae-miR396-5p, tae-miR398, tae-miR9656-3p, tae-miR9676-5p, and tae-miR9778 were significantly highly expressed in tiller primordia and had greatly differential expression (FC > 32) between tiller primordia and young spikes. Among them, expression fold of tae-miR9676-5p and tae-miR9778 were more than 128 times. On the other hand, tae-miR167a, tae-miR6201, tae-miR9666b-3p, tae-miR9670-3p, and tae-miR9777 were highly expressed in young spikes (FC > 4), but only tae-miR6201 had a higher expression fold (FC > 8). Other novel miRNAs showed the same pattern of the expressions as the known miRNAs. DE miRNAs were mainly enriched in tiller primordia and had a higher expression FC compared to young spikes. It was suggested that during the transition from vegetative growth to reproductive growth, the target genes regulated by these miRNAs were mostly up-regulated.

### 3.9. Tissue-Specific miRNAs in TPs

A total of 67 miRNAs were specifically expressed in tiller primordia, including 16 known miRNAs and 51 novel miRNAs (Appendix A). Compared to STs and YSs, both FCs of tae-miR156 were more than 32 times, so it was considered to be able to increase tiller number and severely suppress spike differentiation as reported [42]. Similarly, there were one known miRNA (tae-miR9778) and 9 novel miRNAs (Appendix A) had bigger FCs. It implied that these miRNAs played unknown but important roles during wheat tillering. 

### 3.10. Tissue-Specific miRNAs in YSs

A total of 19 miRNAs were specifically expressed in YSs, including five known miRNAs (tae-miR164, tae-miR167a, tae-miR6201, tae-miR9777, and tae-miR9779) and 14 novel miRNAs (Appendix A). In contrast to the tissue-specific miRNAs in tiller primordia, because stem tips were closely connected with young spikes, the screened tissue-specific miRNAs in young spikes had a smaller FCs between the samples (TP-vs.-YS and ST-vs.-YS). Here, two known miRNAs (tae-miR164 and tae-miR167a) and three novel miRNAs had greater tissue-specificity (FC > 4). Because the close positions between young spikes and stem tips, the FCs of the selected specific miRNAs were smaller than those in tiller primordia. Importantly, the FCs (FC > 3.6) of tae-miR167a, a famous known miRNA, in stem tips and young spikes were the biggest. Previous research demonstrated that overexpression miR167a played vital roles in controlling growth and development of vegetative and flower organs in dicots [72]. So, we speculated that the high expression of these miRNAs affected the development of flower organs at the early stage of wheat spike differentiation. 

### 3.11. Target Relationship Between DE miRNAs and DE mRNAs in Tiller Primordia and Young Spikes

In most cases, the negative correlations of the expression levels between miRNAs and their target mRNAs are often considered as proofs of miRNA targeting [73]. A total of 9788 miRNA-mRNA interaction pairs between tiller primordia and young spikes were discovered by integrative analysis of the mRNA and miRNA data. There were 651 pairs of negative miRNA–mRNA interactions including 63 DE miRNAs (FC > 4) and 416 DE mRNAs in total (Appendix A). 372 negative miRNA–mRNA interaction pairs led to high expression of the target genes in tiller primordia (Figure 6, Appendix A). 279 negative miRNA-mRNA interaction pairs led to high expression of the target genes in young spikes (Figure 7, Appendix A).

Recent reports suggest that some regulations of mRNAs were due to the synergistic and antagonist actions of TFs and miRNAs [74]. Similarly, a total of 81 transcription factor genes were found, including the AP2, ARF, and TCP families. Importantly, growth-regulating factor (GRF) appeared most frequently. Here, we listed some key miRNAs and their targets identified in the integrative regulation network in Table 2.

### 3.12. Expression Profiles of Nine Significant DEGs in Wheat Tiller Primordia, Stem Tips and Young Spikes

Nine significant DEGs were randomly selected on which to perform real-time qRT-PCR (Figure 8). The results showed that the changes of the DEGs at the early developmental stages of wheat, especially at the first two time points, were consistent with the sequencing results. In order to further demonstrate the reliability of the sequencing results, qRT-PCR was performed independently with the same samples used for sequencing (Figure 9). The results were highly consistent.

## 4. Discussion

### 4.1. Typical Expressed Genes During Wheat Tillering and Spike Differentiating

The ribosome-related genes were significantly highly expressed when wheat tillering. All kinds of aquaporin genes, homolog of actin-depolymerizing factor 3 (*ADF3*), some genes related to plant hormones were highly expressed in tiller primordia, including auxin (homologs of *IAA14, IAA15, IAA31, ARF8, ARF13, ARF18*), grbberellin (homologs of *GASA3, GASA4, GASA6, GASA8*), ethylene (homologs of *ERF2, CRF1, CRF4, ERF008, ERF043*), and cytokinin (homolog of *LOGL5*) (Appendix A). 

Water transport across the membranes of various organisms is facilitated by aquaporins [75]. Aquaporin genes can be divided into two major subfamilies, *nodulin 26-like intrinsic proteins* (*NIPs*) and *plasma membrane intrinsic proteins* (*PIPs*), including *NIP1-1, NIP2-2, NIP3-1, PIP1-5,* and *PIP2-5*. Homologs of *NIP1-1, NIP3-1* are highly expressed in root tissue in the face of various abiotic stresses [76,77]. Overexpression of *PIP2-5* aquaporin can alleviate short-term effects of cold soils or low temperature [78]. Plant IAA, ARF, ADF, and PIP family genes are regulators in growth and development, such as lateral root formation and elongation in Arabidopsis and rice, most are associated with auxin signaling [79,80,81]. This study suggested that various IAA, ARF, ADF, GASA, LOGL, ERF, and PIP family genes played important roles during wheat tillering. Obviously, aquaporins and plant hormones played more important roles, particularly auxin. 

The genes related to photosynthesis antenna proteins were most abundant in stem tips, which indicated that antenna systems of plants were particularly important in development of stem tips. Antenna systems of plants are made up of pigment protein complexes belonging to the light harvesting complex (LHC) multigene family [82]. Here, many genes involving photosynthetic systems were homologous to those found in barley, such as *PSAF, PSAH, PSAL,* and *PSBY*. All these genes were highly expressed in stem tips. So, it’s speculated that accumulation of antenna proteins at the stem tips provides sufficient energy and material accumulation for the development of wheat spikes. Additionally, some genes associated with salicylic, jasmonate, and other plant hormone signal transductions were highly expressed in stem tips. For example, the expression level of jasmonate O-methyltransferase gene was very higher in stem tips (Figure 8H). In summary, photosynthesis is the major active metabolism in STs. 

Many typical spike-specific TFs, wheat homologs of *OsMAD32*, *OsMADS55,* and *OsMADS56* were highly expressed in young spikes. For example, *OsMADS32* is a specific MADS-box gene that plays an important role in regulating floral meristem and organs identity [83]. *OsMADS55* is a kind of floret meristem identity gene, facilitating the spikelet to floret meristem transition [84]. *OsMADS56* may form a complex that regulates downstream target genes, overexpression of *OsMADS56* resulted in delayed flowering under long day (LD) [85]. At the same time, auxin (homologs of *ARF2, ARF3, ARF4, ARF11, ARF14, ARF19, IAA16, LAX2*), grbberellin (homologs of *GA2OX1, GA2OX2, GA2OX8*), ethylene (homologs of *ERF12, ESR2, ANT, ERF073, ERF086, EIL3, AIL1, AIL3, AIL5, RAP2-11*), abscisic acid (homologs of *AI5L2, AI5L3, AI5L5*), and cytokinin (homologs of *LOGL9*) related genes were highly expressed in young spikes (Appendix A). Spatiotemporal expression profile analysis demonstrated that *ARF11* was highly expressed in young spikes (Figure 8C). Highly expressed *ethylene-responsive transcription factor 12* (*ERF12*) could play an important role as transcriptional repressor. *ERF* repressors modulate gene expression might have a wide application for crop improvement, particularly in the field of environmental stress tolerance [86]. *Auxin response factor 19* (*OsARF19*) is critical for floral organ development in rice, and enriched in floral and vegetative organ development, especially in young panicles and basal internodes [87]. In addition, the interaction between an ARF and a bHLH transcription factor is biological significant in regulating petal growth, with local auxin levels likely influencing such a biological function [88]. It’s clear that the spike specific TFs, various phytohormone metabolism and signaling genes played important roles when wheat spike differentiation. *FLOWERING LOCUS T (FT)-like* genes control different reproductive traits or stages, *TaFT3* is not found in early young spikes. *FT3* may control spikelet initiation but not floral development in wheat like it does in barley [89]. The most hormone related genes were highly expressed during early spike differentiating, including auxin, grbberellin, ethylene, abscisic acid, and cytokinin, but the highly expressed genes and alleles are different for each hormone, such as highly expressed auxin related genes during tillering are *IAA14, IAA15, IAA31, ARF8, ARF13, ARF18,* those during early spike differentiating are *IAA16, ARF2, ARF3, ARF4, ARF11, ARF14, ARF19, LAX2*. The sophisticated phytohormone regulation mechanism during tillering and spike differentiating in wheat need systemically study.

### 4.2. Typical Expressed miRNAs During Wheat Tillering and Spike Differentiating

The typical expressed miRNAs were identified during wheat tillering and spike differentiating. There are 67 miRNAs typically expressed during tillering, including known wheat miRNAs of tae-miR156, tae-miR171, tae-miR396, and tae-miR398; and 19 miRNAs typically expressed during spike differentiating, including known wheat miRNAs of tae-miR164, tae-miR167, tae-miR6201, and tae-miR9777. Obviously, the most tissue typically expressed miRNAs during wheat tillering and spike differentiating were novel (75.58%). 

There are many reports about miR156, which defines an endogenous flowering pathway by regulating SPL transcription factors [49]. Overexpressing miR156 produces more lateral roots whereas reducing miR156 levels leads to fewer lateral roots in Arabidopsis [41]. The wheat tae-miR156 was highly expressed in all the three tissues, but significantly down-regulated (>5-fold) in stem tips and young spikes. We presumed that tae-miR156 in tiller primordia was closely related to tillering ability in wheat. The miR398 is a typical stresses responsive miRNA, which is proposed to be directly linked to the plant stress regulatory network regulating plant responses to various stress tolerance [90]. The miR396 interacting with growth-regulating factors (GRFs) and *PLETHORA* (*PLTs*) are required for the transition of stem cells into transit-amplifying cells in Arabidopsis [91]. The miR171 over-expression alters the vegetative to reproductive phase transition by activating the miR156 pathway and repressing the expression of the *TRD* (*THIRD OUTER GLUME*) and *HvPLA1* (*Plastochron1*) genes in barley [92]. miR164 and miR167 specify particular cell types during later stages of flower development [93], their homologs tae-miR164 and tae-miR167 were significantly up-regulated in young spikes, implying similar functions. Most functions of these miRNAs were consistent with the tissue-specific miRNAs found here. Because the abundance of miRNA isoforms might have functional consequences on the post-transcriptional regulation of new mRNA isoform targets in different organs [94], these tissue-specific miRNAs imply their extremely important roles in regulating wheat tillering or spike differentiation. However, most tissue-specific miRNAs identified here are novel, how do they regulate wheat tillering and spike differentiation should be systemically studied in the future. 

### 4.3. Phytohormones and TFs Played Key Roles During Wheat Tillering and Spike Differentiating 

The six gene co-expression clusters included many specific TFs and phytohormones. The genes involving in auxin and ethylene metabolism and signaling were almost present in all clusters, indicating the two kind hormones were vitally important for wheat growth and development. But the homologs of the same genes were different among the six clusters, such as the homologs of auxin-responsive protein gene *IAA* described above. According to the gene enrichment referring to KEGG database (Appendix A), plant hormones played different roles during wheat development (Clusters 1, 3, and 6). IAA accumulation is correlated with the number of lateral roots and their primordia [95]. In cluster 6, the genes involving in abscisic acid were highly expressed to promote spike differentiation, as well as the homolog of *abscisic acid-insensitive 5* (*ABI5*) (Appendix A). Arabidopsis ABI5 promoter is active in vegetative and floral tissue, and ABA can induce its expression in specific tissues at later stages [96]. In cluster 3, the genes related to brassinosteroid (BR) were significantly highly expressed (Appendix A). The BRs are essential for plant development and growth in dicot plants, as well as in monocot plants as rice and barley, it regulates leaf, culm development and photomorphogenesis in rice [97]. Our data suggest that all kind hormones are present in the three investigated tissues, but the expressing allelic genes and expression levels of each hormone metabolism are various, which result in tissue typical hormone ingredients. 

On the other hand, all kind of TFs were enriched in each cluster, and there were the most TFs in tiller primordia (clusters K1 and K2; Appendix A). It was speculated that this might be due to the more active differentiation of cell meristem. Tiller primordia had a variety of genes in the bHLH family, such as *bHLH25, bHLH35, bHLH111*, and *bHLH112*, and had many homolog genes of NAC, MYB, WRKY, TCP, and ERF families implying the important roles in the tiller differentiation. At the same time, a number of TF genes involving flower organ development were expressed in young spikes, including homolog genes of WRKY, WOX, TCP, MYB, MADS, HD-ZIP, ARF, and NAC families (clusters K4 and K6;Appendix A). Although transcription regulator activity was not enriched through analysis referring to GO database, many homologs of the TFs regulate florescence development in model plants, such as TCPs and MADS-box TFs. In addition, different genes in MADS family, mainly homeologous to *MADS5, MADS15, MADS34*, and *MADS56* were specifically expressed and played important roles during early spike differentiation. TF is a kind of important regulators in wheat development, but its function is largely unknown. Current research indicates *NAC* and *ERF* are highly expressed in hybrid Jingmai 8, which is related to wheat heterosis [98]. 

Our study suggests DEGs of TF families WOX, B3, Dof, GRF, LBD, and MADS affect the tillering of a dwarf-monoculm wheat mutant *dmc* [6]. The expressions of these TF genes were also fluctuated in the three tissues in this study (Appendix A). Many TF genes associated with wheat tillering or early spike differentiating were identified. Their regulation mechanism should be carefully studied further. 

### 4.4. Post-Transcriptional Regulation of Some Key miRNAs

The tae-miR319 is a homolog of the important post-transcriptional regulator against *TCP* genes in many species. *TCP* genes encode plant-specific transcription factors that regulate plant growth and development by controlling cell proliferation [99]. Wheat tiller primordia are rapidly differentiating meristem groups. Here, the homologous genes belonging to TCP families were found, including the homologs of *OsPCF5, OsPCF6, OsPCF7,* and *OsPCF8*. It showed that these *TCP* genes in rice had similar expression patterns from vegetative to reproductive growth in wheat. As a target gene of tae-miR167a, *auxin response factor 17* (*ARF17*), represses the expression of *GH3* genes and negatively regulates adventitious root formation [100]. It was predicted that miR164 might be involved in the post-transcriptional regulation of homeobox domain, dof domain and helix–loop–helix DNA-binding domain-encoding genes. These kinds of genes are involved in cell-division activating signal and meristem regulation in the root apical meristem [101,102,103]. Then, the integrative analysis about miRNA and the spike development related genes showed that many annotated genes were associated with flower development. Similarly, miR156 targeted SBP domain-encoding genes, these *SPL* genes act to regulate genes mediating cell division, differentiation, and specification in early anther development [44], helping the formation of flowers with highly specific organs. Function annotation suggests that tae-miR9776 regulates expression of transcription factor HBP-1b through plant hormone signal transduction, and plays an important role in biological processes such as floral whorl development and floral organ formation. *OsMADS5*, a MADS box gene, encodes a regulatory protein in flower meristem determining floral organ identity. Previous researches indicate that *OsMADS5* is structurally related to the AGL2 family and may be involved in controlling flowering time [104]. Wheat homolog of *OsMADS5* was found to be regulated by three novel miRNAs. Some important genes, such as *ARFs* and growth-regulating factors (GRFs) were also regulated by several miRNAs.

### 4.5. Regulation Networks of DE mRNA and DE miRNA in Tiller Primordia and Young Spikes

A number of miRNAs have been shown to play important regulatory roles in cell meristem and differentiation in various plants. Finding regulatory mRNA targets is essential to understanding the biological functions of the miRNAs. False positives of miRNA target prediction were reduced by limiting the number of mismatches and increasing the threshold of FC of DE miRNAs. 

In this study, tae-miR156 and tae-miR167a were recognized to play important roles during tillering and spike differentiating. The highly conserved miR156 regulates transcription factor SPLs to define an endogenous flowering pathway in Arabidopsis [49]. Here, *Separase* (*ESP*) was identified as a target gene of miR156, too. According to other reports, ESP plays central role in controlling the release of sister chromatid cohesion during mitosis and meiosis, and transcripts of *ESP* are present at approximately equal levels in roots, stems, leaves, and buds in Arabidopsis [105]. Our results were consistent with these facts, for numerous mitosis cells exist in the meristem of tiller primordia and young spikes. The basic mechanisms in these processes are known. However, the differences maybe exist among various organisms. 

Auxin response factors *ARF2-ARF4* and *ARF5* play significant roles in regulating both female and male gametophyte development in Arabidopsis [106]. Here, homolog of *ARF17* was identified as a new target gene of tae-miR167a in regulatory network. *ARF17* is a target of miR160, and rising mRNA levels of *ARF17* result in dramatic developmental defects, including premature inflorescence development, altered phyllotaxy along the stem, reduced petal size, abnormal stamens, sterility in Arabidopsis [107]. Thus, both miR160 and miR167a may play roles in the post-transcriptional regulation of *ARF17* and affect the stability and regulation of auxin metabolism and signaling. 

In addition to ARF and GRF families, the transcription factors such as WRKY, bZIP, C3H, B3, EIL, G2-like, AP2 were significantly differentially expressed between tiller primordia and young spikes, which involving in the conversion from vegetative growth to reproductive growth. In particular, *ARF11, GRF2, GRF4, GRF10, GRF11, GRF12,* and *GRF14* were co-regulated by at least 10 miRNAs in a regulatory network (Figure 10). 

Then, we established a primary model of the wheat tillering and spike differentiating related key miRNA–mRNA regulation network. Among them, 12 key known miRNAs and 16 novel miRNAs were further analyzed (Figure 10). Here, tae-miR9778 and novel miRNA (AN: uggcugugaugauauaac) involved in the most significant metabolic pathways in tiller primordia and young spikes (Table 1), they are ribosome pathway (ko03010) in tiller primordia and plant hormone signal transduction (ko04075) in young spikes. These data indicated that target genes were regulated by known miRNAs and novel miRNAs. Among these miRNAs, the regulation of novel miRNAs occupy a very large position and are waiting to be studied in the whole regulatory network.

## 5. Conclusions

This study was to investigate and establish wheat standard models of the gene expression profiles and microRNA regulation networks at the tillering and early spike differentiating stages employing Guomai 301, a representative excellent new high yield wheat cultivar in the Henan province in China. We identified a set of DEGs and DE miRNAs in tiller primordia, stem tips, and young spikes of Guomai 301. Six major expression profile clusters of tissue-specific DEGs for the three tissues were classified by gene co-expression analysis, and we established the miRNA–mRNA regulatory networks during tillering and early spike differentiating. Our data indicated that some important target genes, such as the homologs of the members in *ARF* and *GRF* families, were regulated by multiple miRNAs. The model of the wheat tillering and spike differentiating related key miRNA-mRNA regulation network provides a solid background for future systemic studies on wheat tillering and early spike differentiating, as well as molecular breeding.

## Figures and Tables

**Figure 1 genes-10-00686-f001:**
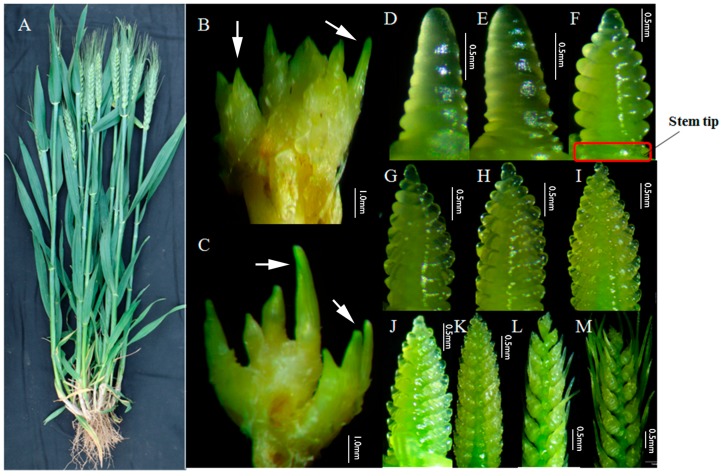
The plant, tillers, and young spikes of Guomai 301. (**A**) An individual plant of Guomai 301 in the field condition; (**B**,**C**) tiller primordia at three-leaf stage and four-leaf stage, arrow heads indicated the small tillers; (**D**–**M**), spikes at various developmental stages; (**D**) single ridge stage; (**E**) double ridge stage; (**F**) glume primordia visible; (**G**) lemma primordia visible; (**H**) floret primordia visible; (**I**) late terminal spikelet; (**J**) terminal spikelet stage; (**K**) two awns/spikelet reaching apical meristem of the spikelet; (**L**) two awns/spikelet twice as long as spikelet; (**M**) elongation of third awn/spikelet, basal floret fully covered by lemma. Stem tip, the stem part connected to the young spikes.

**Figure 2 genes-10-00686-f002:**
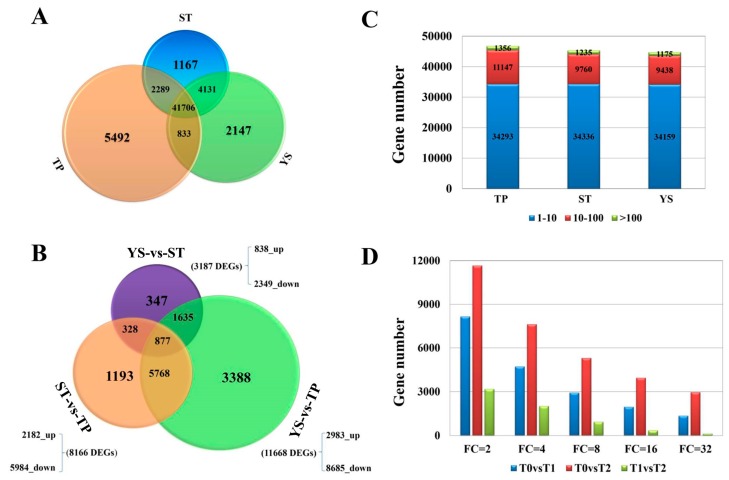
The gene expression profiles in tiller primordia (TP), stem tips (ST), and young spikes (YS) of Guomai 301. (**A**) A Venn diagram showing the numbers of expressed genes in various samples, the FPKM is above 1 in all three replications. (**B**) Statistics of differentially expressed genes (DEGs) in each sample. (**C**) Numbers of genes expressed in each sample with an average number of fragments per kilobase of transcript per million mapped fragments (FPKM) ≥ 1. (**D**) numbers of DEGs with different Fold Changes.

**Figure 3 genes-10-00686-f003:**
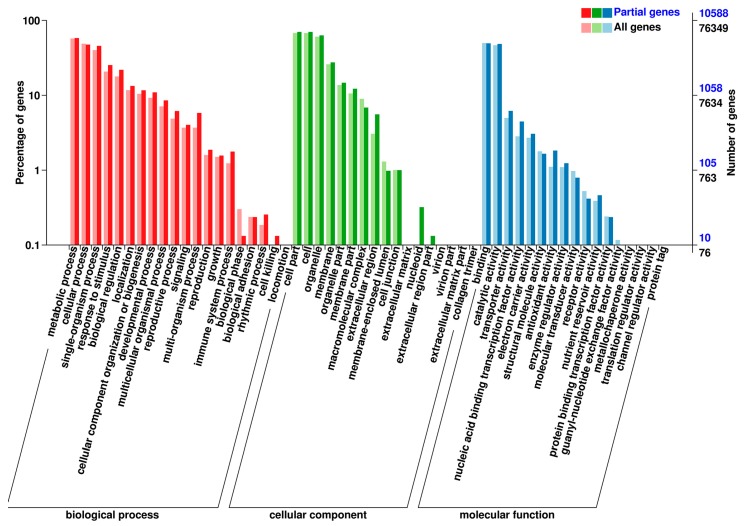
Functional classification of differentially expressed genes (DEGs) between tiller primordia and young spikes in the Gene Ontology (GO) database.

**Figure 4 genes-10-00686-f004:**
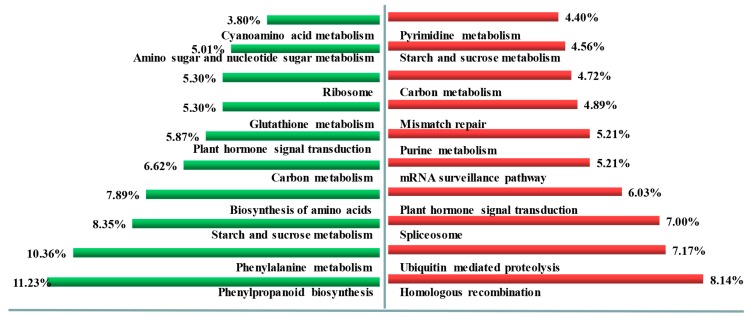
The top ten enhanced pathways in the tiller primordia and young spikes. Left, the enhanced pathways in tiller primordia; Right, the enhanced pathways in young spikes. Percentage, the ratio of the number of DEGs annotated to one pathway to the number of DEGs annotated to all pathways.

**Figure 5 genes-10-00686-f005:**
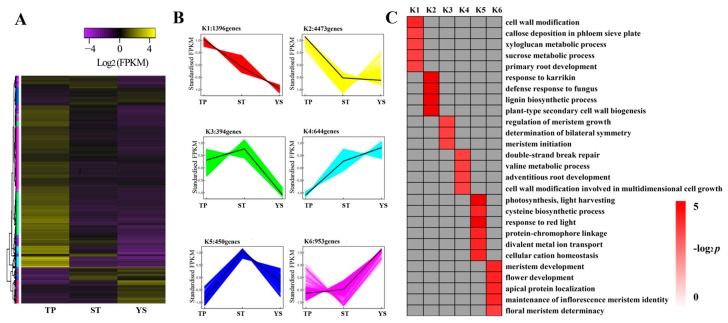
Overview of serial analysis of DEGs identified by pairwise comparisons of TPs (tiller primordia), STs (stem tips), and YSs (young spikes). (**A**) a heatmap of DEGs across TP, ST, and YS. Expression values of the three tissues were presented as log_2_-transformed normalized FPKM values. Six clusters were showed. (**B**) K1–K6, the six clustered DEGs in the three tissues. (**C**) GO-term function enrichment of the six clusters. The significances of the most represented GO-terms in each main cluster were indicated using log_2_-transformed *p*-value (red). The dark grey areas represented the missing values.

**Figure 6 genes-10-00686-f006:**
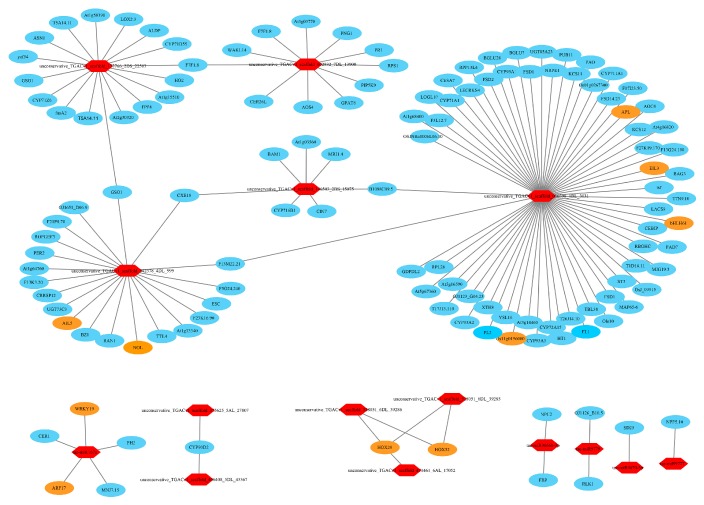
A part of the 372 negative microRNA (miRNA)–mRNA interaction pairs (regulation network) in tiller primordia. Red hexagons represented the highly expressed miRNAs in tiller primordia. Ovals represented the target genes of miRNAs. Yellow ovals represented transcription factors (TFs). Higher resolution image is shown in Appendix A.

**Figure 7 genes-10-00686-f007:**
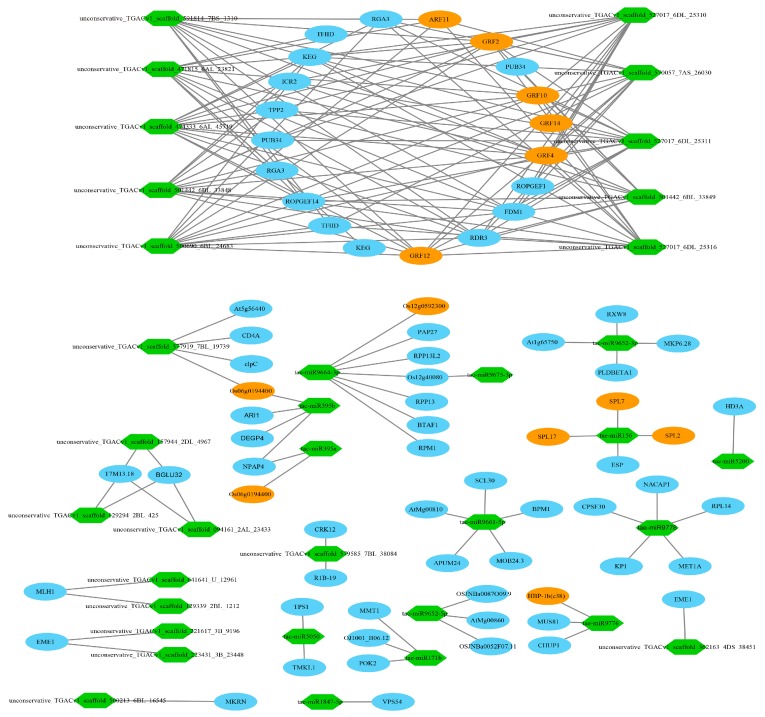
A part of the 279 negative miRNA–mRNA interaction pairs (regulation network) in young spikes. Green hexagons represented the highly expressed miRNAs in young spikes. Ovals represented the target genes of miRNAs. Yellow ovals represented TFs. Higher resolution image is shown in Appendix A.

**Figure 8 genes-10-00686-f008:**
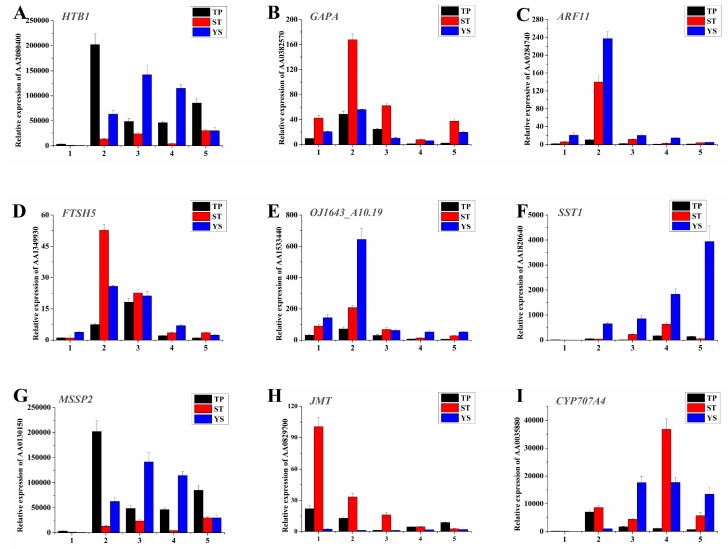
Spatiotemporal expression profiles of the nine DEGs in TPs (tiller primordia), STs (stem tips), and YSs (young spikes). (**A**) AA2080400 (Histone H2B.1); (**B**) AA0382570 (Glyceraldehyde-3-phosphate dehydrogenase A, chloroplastic); (**C**) AA0284740 (Auxin response factor 11); (**D**) AA1349930 (ATP-dependent zinc metalloprotease FTSH 5, mitochondrial); (**E**) AA1533440 (Replication protein A 32 kDa subunit B); (**F**) AA1820640 (Sucrose: sucrose 1-fructosyltransferase); (**G**) AA0130150 (Monosaccharide-sensing protein 2); (**H**) AA0829700 (Jasmonate O-methyltransferase); (**I**) AA0035880 (Abscisic acid 8 and apos;-hydroxylase 4). The *actin* gene was used as internal control. The number 1–5 of *x*-axis indicated the sampling dates, and functional annotation and the other details of A-I were listed in Appendix A. All quantitative reverse transcription PCR (qRT-PCR) reactions were replicated three times.

**Figure 9 genes-10-00686-f009:**
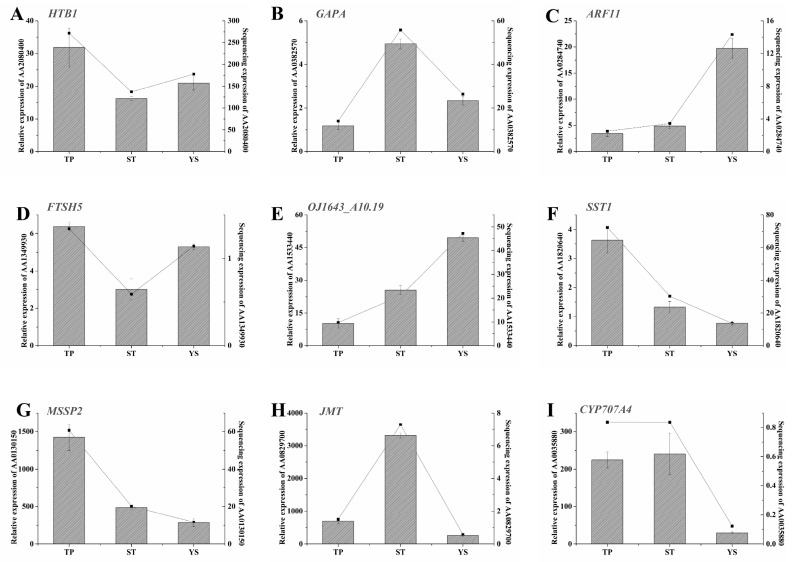
Spatiotemporal expression profiles of the nine DEGs in TPs (tiller primordia), STs (stem tips), and YSs (young spikes). (**A**) AA2080400 (Histone H2B.1); (**B**) AA0382570 (Glyceraldehyde-3-phosphate dehydrogenase A, chloroplastic); (**C**) AA0284740 (Auxin response factor 11); (**D**) AA1349930 (ATP-dependent zinc metalloprotease FTSH5, mitochondrial); (**E**) AA1533440 (Replication protein A 32 kDa subunit B); (**F**) AA1820640 (Sucrose: sucrose 1-fructosyltransferase); (**G**) AA0130150 (Monosaccharide-sensing protein 2); (**H**) AA0829700 (Jasmonate O-methyltransferase); (**I**) AA0035880 (Abscisic acid 8 and apos;-hydroxylase 4). Functional annotation and the other details of A-I were listed in Appendix A. The *actin* gene was used as internal control. Left *y*-axis represented relative expression. Relative expressions were showed by histograms. Error bars indicated the standard deviation. Right *y*-axis represented expression value (FPKM) of transcriptome sequencing. Expression value was replaced by the average of three repetitions and showed by line charts. *x*-axis, the samples used for sequencing. All quantitative reverse transcription PCR (qRT-PCR) reactions were replicated three times.

**Figure 10 genes-10-00686-f010:**
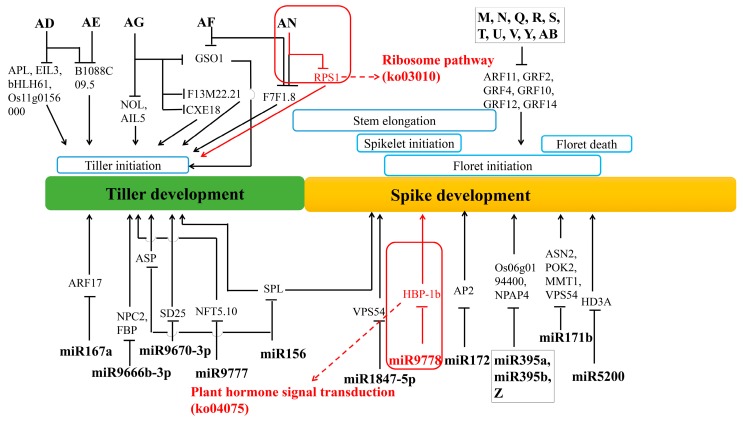
A model of the wheat tillering and spike differentiating related key miRNA–mRNA regulation network. The red rectangles indicate the miRNA and targets involved in the two most significant metabolic pathways in tiller primordia and young spikes (Table 1). For short the “tae-” before the known miRNAs is omitted. Words A-AN represented different novel miRNAs, the original ID corresponding to these new miRNAs were listed in Appendix A.

**Table 1 genes-10-00686-t001:** The major pathways in the three tissues referring to Kyoto Encyclopedia of Genes and Genomes (KEGG) database.

Tissue	Pathway	Ko id	*p*-Value	Corrected *p*-Value
Tiller primordia	Ribosome	ko03010	3.24 × 10^−13^	3.34 × 10^−11^
	Biosynthesis of amino acids	ko01230	2.83 × 10^−7^	2.91 × 10^−5^
	Nitrogen metabolism	ko00910	3.28 × 10^−7^	3.38 × 10^−5^
Stem tips	Photosynthesis - antenna proteins	ko00196	0	0
	Carbon fixation in photosynthesis	ko00710	2.84 × 10^−11^	1.30 × 10^−9^
	Carbon metabolism	ko01200	1.52 × 10^−6^	6.98 × 10^−5^
Young spikes	Plant hormone signal transduction	ko04075	3.03 × 10^−5^	0
	DNA replication	ko03030	0.01	0.23

**Table 2 genes-10-00686-t002:** Some key miRNAs and their targets identified in the integrative regulation network.

miRNA ID ^1^	miRNA Sequence	log_2_FC	Target Genes or Transcription Factor Genes ^2^
miR156	TGACAGAAGAGAGTGAGCACA	−5.9	*ESP*, ***SPL2***, ***SPL7***, ***SPL17***
miR167a	TGAAGCTGCCAGCATGATCTA	2.6	*CER1*, *FH2*, ***WRKY19***, ***ARF17***, *MNJ7.15*
miR9776	TTGGACGAGGATGTGCAACTG	−3.61	*MUS81*, *CHUP1*, ***HBP-1b(c38)***
M	UUCCACAGCUUUCUUGAACUG	−3.76	***GRF12***, ***ARF11***, *RDR3*, *FDM1*, *KEG*, ***GRF4***, *ROPGEF14*, ***GRF10***, ***GRF2***, *RGA3*, *TFIID*
N	UUCCACAGCUUUCUUGAACUU	−3.7	*TPP2*, *ICR2*, ***GRF12***, *RDR3*, *FDM1*, *KEG*, ***GRF4***, *ROPGEF14*, *PUB34*, ***GRF10***, ***GRF2***, *RGA3*
P	UGUUCUGAAGAAACUGUCACC	−2.19	***MKRN***
Q	UUCCACAGCUUUCUUGAACUG	−3.76	***GRF12***, ***ARF11***, *RDR3*, *FDM1*, *KEG*, ***GRF4***, *ROPGEF14*, ***GRF10***, ***GRF2***, *RGA3*, *TFIID*
R	UUCCACAGCUUUCUUGAACUU	−3.7	*TPP2*, *ICR2*, ***GRF12***, *RDR3*, *FDM1*, *KEG*, ***GRF4***, *ROPGEF14*, *PUB34*, ***GRF10***, ***GRF2***, *RGA3*
S	UUCCACAGCUUUCUUGAACUU	−3.7	*TPP2*, *ICR2*, ***GRF12***, *RDR3*, *FDM1*, *KEG*, ***GRF4***, *ROPGEF14*, *PUB34*, ***GRF10***, ***GRF2***
T	UUCCACAGCUUUCUUGAACUU	−3.7	*TPP2*, *ICR2*, ***GRF12***, *RDR3*, *FDM1*, *KEG*, ***GRF14***, *ROPGEF1*, *PUB34*, ***GRF10***, ***GRF4***, *GRF2*
U	UUCCACAGCUUUCUUGAACUU	−3.7	*TPP2*, *ICR2*, ***GRF12***, *RDR3*, *FDM1*, *KEG*, ***GRF4***, *ROPGEF14*, *PUB34*, ***GRF10***, ***GRF2***, *RGA3*
V	UUCCACAGCUUUCUUGAACUG	−3.76	***GRF12***, ***ARF11***, *RDR3*, *FDM1*, *KEG*, ***GRF4***, *ROPGEF14*, ***GRF10***, ***GRF2***, *RGA3*, *TFIID*
Y	UUCCACAGCUUUCUUGAACUG	−3.76	***GRF12***, ***ARF11***, *RDR3*, *FDM1*, *KEG*, ***GRF4***, *ROPGEF14*, ***GRF10***, ***GRF2***, *RGA3*, *TFIID*
AD	CCUGUUGAGCUUGACCCC	2.41	*CESA7*, *CYP71A1*, *LOGL10*, *F3L12.7*, *At1g68400*, *OSJNBa0006L06.10*, *GDPDL2*, *RPL28*, *At5g67360*, *At5g16590*, *T17J13.110*, *OJ1123_G04.23*, *F13M22.21*, *CYP93A2*, *XTH8*, *FL2*, *YSL16*, ***Os11g0156000***, *At3g14460*, *CYP93A3*, *CYP72A15*, *HT1*, *T26J14.10*, *FL1*, *TBL38*, *Ole10*, *FSD1*, *MAP65-6*, *ST3*, *OsJ_09915*, *T1D16.11*, *B1088C09.5*, *MJG19.3*, *RBOHC*, *FAD7*, *CEBIP*, ***bHLH61***, *LACS9*, *T7N9.10*, *tsf*, *BAG3*, ***EIL3***, *F13G24.190*, *F27K19.170*, *At4g16820*, *KCS12*, *AOC4*, ***APL***, *F17I23.50*, *F5D14.23*, *CYP711A1*, *Os01g0267300*, *PAO*, *KCS11*, *PUB11*, *NRPE1*, *UGT85A23*, *FSD1*, *BGLU7*, *CYP93A1*, *BGLU28*, *FSD2*, *RPP13L4*, *LECRKS4*
AJ	UCGGACCAGGCUUCAUUCCUU	2.02	***HOX29***
AL	UCGGACCAGGCUUCAUUCCUU	2.02	***HOX29*, *HOX32***
AM	UCGGACCAGGCUUCAUUCCUU	2.02	***HOX29*, *HOX32***

^1^ Words A-AN in miRNA ID represented different novel miRNAs, the original ID corresponding to these new miRNAs were listed in Appendix A. ^2^ The transcription factor genes were highlighted in bold type, target genes were named referring to Swiss-Prot database (https://www.uniprot.org). For short the “tae-“ before the known miRNAs is omitted.

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
