# Peer review of "Gene Expression Profiles and microRNA Regulation Networks in Tiller Primordia, Stem Tips, and Young Spikes of Wheat Guomai 301"

_genes, 2019, doi:10.3390/genes10090686_

Round 1
Reviewer 1 Report
The manuscript entitled ‘Gene expression profiles and microRNA regulation 2 networks in tiller primordia, stem tips and young 3 spikes of wheat Guomai 301’ described an RNA sequencing approach of the high-yielding wheat cultivar Guomai 301. Since grain yield depends on a number of tillers, numbers of grains per spike and other factors the authors concentrated on three developmental relevant tissues, namely tiller primordia, stem tips and young spikes. The data are interesting and the experiments were carefully performed and the data properly presented. But I have some comments which should be considered before publication.
In general, keywords should not repeat the title. At least four of the six keywords need to be replaced. In addition, the manuscript contains some typos which I did not list here.
Although quite some literature was cited from my point of view some of the recent but important literature should be discussed and included. In the end, I will provide a list of the details.
Another important advice I want to give is to omit the majority of abbreviations for the three tissues (TP, ST and YS). Because of extensive use, the manuscript is hardly readable. It really will improve dramatically if the authors follow this comment.
I don’t like that the majority of data has been transferred to the supplement. Since Genes has no limitation in a number of figures, I would request to include the main data in the main text.
Some minor aspects:
L37 – the more common term is thousand-grain weight instead of kilo-grain weight.
L204 – replace ‘has many grains’ with the real numbers, for example between 20-24 grains.
Figure 2 caption, since all the graphs are using abbreviations the authors must include the description of the abbreviations, too.
Figures 8 and 9: In order to make it easier for the reader I suggest to integrate the gene names into each diagram rather than mentioned in the captions.
In the caption of Figure 9, it becomes not clear that the samples used for qRT-PCR are the same as used for sequencing. Please indicate this in the revised version.
In summary, the manuscript would profit from a revision.
Koppolu and Schnurbusch (2019) Developmental pathways for shaping spike inflorescence architecture in barley and wheat. J Integr Plant Biol 61: 278-295
Sakuma et al (2019) Unleashing floret fertility in wheat through the mutation of a homeobox gene. Proc Natl Acad Sci U S A 116: 5182-5187
Wolde et al (2019) Genetic modification of spikelet arrangement in wheat increases grain number without significantly affecting grain weight. Mol Genet Genomics 294: 457-468
Guo et al (2018a) Manipulation and prediction of spike morphology traits for the improvement of grain yield in wheat. Sci Rep 8: 14435
Guo et al (2018b) Plant and floret growth at distinct developmental stages during the stem elongation phase in wheat. Front Plant Sci 9: 330
Wang et al (2018) Abscisic acid influences tillering by modulation of strigolactones in barley. J Exp Bot 69: 3883-3898
Mulki et al (2018) FLOWERING LOCUS T3 control spikelet initiation but not floral development. Plant Physiol. 178 (3) 1170-1186
Reviewer 2 Report
The presented study investigate wheat gene expression profiles and microRNA regulatory networks within the Guomai 301. The authors identified a set of differentially expressed miRNAs in three tissues in Guomai 301. The authors also explore the miRNA-mRNA interaction network and suggest that some genes are regulated by multiple miRNAs. However, the authors should address some points:
1.miRNA datasets method profiling should be better explained. Why authors decided to use SILVA database which is preferably used for metagenomics and contain predominantly bacterial 16s. Which DB is used for snRNA, ncRNAs,snoRNA? What mapping parameters are used for Bowtie, mismatches, etc and why?
2. The Paragraph "2.6. Prediction of the miRNA target genes" is poorly described. miRNAfinder is a pre-miRNA prediction tool. How exactly the authors use the specific tool for target gene prediction as this feature was not described in the software?
3. A 3'RACE is needed for the most important regulatory module genes that authors suggest to confirm gene targeting.
